# Tissue Microarray Lipidomic Imaging Mass Spectrometry Method: Application to the Study of Alcohol-Related White Matter Neurodegeneration

Isabel Gameiro-Ros [1,†], Lelia Noble [2,†], Ming Tong [3,†], Emine B. Yalcin [2] and Suzanne M. de la Monte [2,3,4,*]

1   Department of Pharmacology and Therapeutics, Faculty of Medicine, Autonomous University of Madrid, 28029 Madrid, Spain
2   Department of Pathology and Laboratory Medicine, Rhode Island Hospital, Alpert Medical School of Brown University, Providence, RI 02903, USA
3   Department of Medicine, Rhode Island Hospital, Alpert Medical School of Brown University, Providence, RI 02903, USA
4   Departments of Neurology & Neurosurgery, Rhode Island Hospital, Alpert Medical School of Brown University, Providence, RI 02903, USA
*   Correspondence: suzanne_delamonte_md@brown.edu; Tel.: +1-401-444-7364
†   These authors contributed equally to this work.

**Abstract:** Central nervous system (CNS) white matter pathologies accompany many diseases across the lifespan, yet their biochemical bases, mechanisms, and consequences have remained poorly understood due to the complexity of myelin lipid-based research. However, recent advances in matrix-assisted laser desorption/ionization-imaging mass spectrometry (MALDI-IMS) have minimized or eliminated many technical challenges that previously limited progress in CNS disease-based lipidomic research. MALDI-IMS can be used for lipid identification, semi-quantification, and the refined interpretation of histopathology. The present work illustrates the use of tissue micro-arrays (TMAs) for MALDI-IMS analysis of frontal lobe white matter biochemical lipidomic pathology in an experimental rat model of chronic ethanol feeding. The use of TMAs combines workload efficiency with the robustness and uniformity of data acquisition. The methods described for generating TMAs enable simultaneous comparisons of lipid profiles across multiple samples under identical conditions. With the methods described, we demonstrate significant reductions in phosphatidylinositol and increases in phosphatidylcholine in the frontal white matter of chronic ethanol-fed rats. Together with the use of a novel rapid peak alignment protocol, this approach facilitates reliable inter- and intra-group comparisons of MALDI-IMS data from experimental models and could be extended to human disease states, including using archival specimens.

**Keywords:** MALDI; white matter; lipidomic; alcohol; tissue micro-array; mass spectrometry; central nervous system

## 1. Introduction

### 1.1. White Matter Pathology in Neurodegeneration

White matter (WM) atrophy is an important and consistent feature of many chronic central nervous system (CNS) diseases, including alcohol-related brain degeneration (ARBD) [1–3], Alzheimer's disease (AD) [4–6], vascular dementias [7,8], and frontotemporal lobar degeneration [9], yet the mechanisms have been vastly under-investigated due to technical challenges posed by myelin lipidomic research. WM is mainly composed of axons that project neuronal connections to different brain regions and myelin, which provides the insulation and support needed to optimize axonal functions. Damage to or the loss of axons disrupts structural and pathway communications. The degeneration or loss of myelin compromises the efficiency of neurotransmission needed for cognitive and motor

functions [10–17] and renders axons vulnerable to toxic, metabolic, and inflammatory injury that can lead to sustained or permanent functional deficits [18].

*1.2. Alterations in Myelin Lipid Composition with Disease*

A unique property of CNS myelin is its very high dry mass of lipid (70–85%) compared with that of protein (15–30%). Major WM lipids include cholesterol, glycosphingolipids, sulfatides, gangliosides, phospholipids, and sphingomyelin [19]. Chronic disease states leading to WM degeneration are marked by altered expression and metabolism of phospholipids and sulfatides [20–22]. Alterations in membrane phospholipid composition perturb lipid raft and receptor functions [23,24]. The aberrant or reduced expression of glycerosphingolipids, sphingomyelins, and sulfatides broadly impairs neuronal and glial functions, including plasticity, neuronal conductivity, memory, myelin maintenance, protein trafficking, adhesion, glial-axonal signaling, insulin secretion, and oligodendrocyte survival [20,25,26]. The degradation of sulfatide increases ceramide abundance [27,28], which promotes neuroinflammation, reactive oxygen species formation, and apoptosis, and impairs cellular survival and metabolic signaling [25].

*1.3. Strategy for Increasing Knowledge of Disease-Specific WM Pathology*

Although imbalances in phospholipids or sphingolipids represent biochemical correlates of WM degeneration, the breadth of knowledge in this field is limited. Thus far, relatively few lipids have been characterized in relation to disease, and the available information about the specificity, time course, and reversibility of CNS myelin lipid pathology is modest. Progress in this field has been impeded by the lack of accessible tools for the simultaneous analysis of multiple samples, including different sources, such as diseased versus control brains, and correlations of biochemical parameters with histopathology. Herein, we describe the use of tissue microarrays (TMAs) coupled with matrix-assisted laser desorption/ionization-imaging mass spectrometry (MALDI-IMS) to characterize the effects of ARBD on frontal lobe WM lipid profiles in an established experimental rat model of chronic ethanol consumption.

*1.4. Benefits of Tissue Microarrays in Research*

TMAs enable the simultaneous in situ analysis of multiple tissue samples for correlations with histopathology and intra- and inter-group comparisons. The TMA approach has been applied to studies of brain microvascular pathology in neurodegeneration [29], gene expression changes in malignancies [30,31], and proteomic mass spectrometry [32]. The unique feature of this report is the utilization of TMA technology for untargeted disease-oriented lipidomic research. Our methodological approach was designed to overcome limitations of data acquisition and interpretation encountered with standard protocols that are not focused on lipidomics research. Furthermore, although we have successfully utilized standard MALDI-IMS procedures [33–35], the need to ensure the uniform handling and simultaneous analysis of samples to soundly investigate the pathophysiology of diseases, either experimental or human, drew us to re-strategize by developing a TMA-MALDI-Lipidomics protocol. The only currently feasible alternative would be to extract lipids for the simultaneous analysis of samples deposited onto a MALDI target plate [36], but that approach abrogates correlative in situ studies of tissue pathology.

## 2. Materials and Methods
*2.1. Overview*

The methods herein detail our TMA-MALDI-Lipidomics protocol, including the critical reagents and procedural steps needed to generate TMAs that are suitable for lipidomic research. The example illustrated is an established experimental rat model of chronic alcohol exposure that causes WM degeneration together with deficits in spatial learning and memory tasks as demonstrated using the Morris Water Maze [37]. This section includes a brief description of the model and a summary of the technical details pertaining to the

acquisition of MALDI-IMS data. The methods corresponding to both the rat model and MALDI-IMS have been detailed in earlier publications [21,38,39].

*2.2. Experimental Model*

Chronic ethanol feeding of adult Long Evans rats causes significant deficits in spatial learning and memory, along with atrophy and degeneration of WM [38,40,41]. To generate the model, 4-week-old Long Evans rats were pair-fed for 6 weeks with commercial isocaloric liquid diets (Research Diets, Inc., New Brunswick, NJ, USA) that contained 0% or 36% (caloric content) ethanol [21,42]. The diets were prepared daily according to the manufacturer's protocol. Immediately after sacrifice via isoflurane inhalation, the brains were harvested to obtain a standardized slice of both frontal lobes by making an initial cut just anterior to the temporal tips and a second cut 3 mm further anterior. The right and left 3 mm-thick frontal lobe slices were laid flat in separate pre-labeled Tissue-Tek acetyl polymer square mesh slotted embedding and processing cassettes (Electron Microscopy Sciences, Hatfield, PA, USA). One slice was quickly frozen on dry ice and stored at −80 °C, and the other was immersion-fixed in 10% neutral buffered formalin and stored at 4 °C. The use of experimental animals for this research was approved by the Lifespan Institutional Animal Care and Use Committee, Providence RI, USA.

*2.3. TMA Generation*

2.3.1. Overview

TMAs were generated to enable the batch processing and analysis of multiple samples on a single slide. The TMAs should optimally include 2 or 3 replicate control and experimental samples, randomly positioned to assess the reproducibility of results and perform robust intra- and inter-group comparisons [43,44]. The main considerations for generating TMAs are as follows: (a) nature of the embedding compound; (b) tissue state (fresh-frozen or formalin-fixed); and (c) technical steps used to efficiently collect and transfer samples for constructing the arrays.

2.3.2. Selection of the Embedding Compound

Embedding compounds must enable optimum tissue sectioning and minimize background signals. In addition, they should be easy to cut and withstand repeated cryosectioning. Embedding compounds that generate high backgrounds or diminish the MALDI-IMS signal-to-noise ratios are unsuitable. Since the suitability and effects of embedding compounds can vary with the tissue type, source, and preparation, preliminary studies must be conducted to optimize experimental conditions. The sources and preparations of four embedding compounds considered in developing the MALDI-TMA method are provided below.

a.  Tissue-Plus Optimal Cutting Temperature embedding compound (Tissue-Tek O.C.T.) from Sakura Finetek USA Inc., Torrance, CA, USA, was purchased and used according to the manufacturer's instructions.
b.  Two percent carboxymethylcellulose (CMC) gel was prepared by dissolving high-viscosity carboxymethlycellulose sodium salt (Sigma-Aldrich, St. Louis, MO, USA) in sterile, deionized water and storing the product at 4 °C for up to 2 weeks.
c.  Modified O.C.T. (mOCT) was prepared as described [45]. In brief, polyvinyl alcohol (PVA) 6–98 (10 g) was heat-solubilized in Hank's Balanced Salt Solution (HBSS; 100 mL), and after cooling to room temperature, polypropylene glycol (PPG) 2000 (8 mL) and sodium azide (100 mg) were added and vortex-mixed to form a thin milky white gel. The mOCT was stored at room temperature for up to 3 months and vortex mixed just prior to each use.

d.  Gelatin (10–15%) is the fourth potential embedding compound that ultimately was not tested for this study [46–48] due to very low signals obtained with WM tissue, as previously reported [46]. Unfortunately, the gelatin problem could not be resolved because it would have been impracticable to precisely peel the embedding compound away from the TMA cores.

### 2.3.3. Tissue Sample Preparation for MALDI-IMS Lipidomic Studies

1.  Only fresh-frozen or formalin-fixed tissues are suitable.
2.  Paraffin-embedded samples are unsuitable because alcohol-containing solvents used for tissue processing destroy and solubilize lipids.
3.  For sample testing, 3 mm-diameter fresh-frozen or formalin-fixed frontal lobe WM tissue cores from adult Long Evans rats (2 control and 2 alcohol-fed) were embedded in O.C.T., CMC, or mOCT.

### 2.3.4. TMA Construction

The workflow diagram in Figure 1 depicts the required steps for generating sample recipient blocks, coring the samples, and assembling the TMAs.

(1) Generating TMA recipient blocks:
- (a)  Plan the array.
  - 1.  Select a validated and reproducible system for making TMA recipient blocks.
    - a.  We used reusable silicone molds from Arraymold (Salt Lake City, UT, USA).
  - 2.  Decide on the number of samples to be included and core diameter requirements for the TMA.
    - a.  Arraymold recipient block template configurations: 1 mm cores = 170 samples; 1.5 mm cores = 150 samples; 2 mm cores = 70 samples; 3 mm cores = 40 samples; 4 mm cores = 15 samples; 5 mm cores = 15 samples.
  - 3.  Plan to include either fresh-frozen or formalin-fixed tissue in a single TMA.
    - a.  Different tissue preparations can impact MALDI-IMS signal intensities.
    - b.  Large differences in MALDI-IMS signal intensities significantly distort results.
  - 4.  Construct an asymmetric grid map to designate sample core insertion sites. Asymmetric mapping:
    - a.  Establishes orientation of the TMAs.
    - b.  Designates sample addresses to prevent misidentification at later analytical stages.
- (b)  Evenly distribute the embedding compound across the mold, then allow the compound to solidify in a cryostat chamber set to −20 °C.
  - 1.  For this protocol, we used a Leica CM 3050 cryostat microtome (Leica Biosystems, Wetzlar, Germany).
- (c)  Carefully release the frozen recipient block by peeling off the silicone mold.
- (d)  Store the recipient block at −80 °C in an air-tight container for up to 3 months. Commercial air-tight laboratory plastic containers were used to store TMA slides organized and held in molded plastic cork-lined slide storage boxes that were further sealed in plastic bags.

(2) Sample preparation and coring:
- (a)  Keep samples in their original cassettes at least until the tissue cores have been harvested and transferred to the recipient block.
- (b)  Select a 1 mm to 5 mm diameter re-usable Arraymold coring tool or a disposable surgical biopsy punch that corresponds to the recipient block.

1. Match the Arraymold coring tool size to the commercial Arraymold to enable snug fitting of the specimens and minimize gaps.

(c) For fresh-frozen tissue, retrieve samples from the −80 °C freezer and equilibrate for 15–30 min in a −20 °C cryostat chamber.

(d) For formalin-fixed tissue, prepare samples 2 or 3 days in advance by thoroughly rinsing them (in their original cassettes) in phosphate buffered saline (PBS) at 4 °C for 48 h with gentle platform agitation.

(3) Assembling the TMA:

(a) Transfer the recipient block from the −80 °C freezer to the cryostat chamber (−20 °C) and equilibrate for 15–30 min before use.

1. Keep the recipient block in the cryostat chamber until all cored samples have been transferred.

(b) According to the grid map, fill designated blank wells with embedding compound only.

(c) For fresh-frozen samples, just prior to generating the cores, semi-thaw the tissue slices, on a bed of wet ice, one by one. Use aluminum foil to separate tissue from wet ice.

1. Obtain the core (gentle pressure may be required) and immediately transfer it to the recipient block taking care to avoid thawing.

2. Use a cryostat-chilled metal spatula to tap the core flush with the recipient block's surface.

(d) For formalin-fixed samples, blot the tissue dry with lint-free laboratory grade paper wipes.

1. Obtain the core and immediately transfer it to the recipient block.

2. Alternatively, core the tissue and momentarily leave it in place but slightly elevated above the surrounding tissue to apply a unique Microdot (1 μL) orientation pattern to the sample edges using surgical biopsy ink (Margin-Marker; Vector Surgical, Waukesha, WI, USA) [33], and then transfer it to the recipient block.

3. Use gentle tapping with a pre-chilled metal spatula to fully insert the core.

(e) To control for reproducibility of the results, include duplicate or triplicate samples spatially dispersed across the TMA.

(f) Carefully fill gaps between tissue cores and well walls with supplemental embedding compound, particularly if the frozen cores fragment.

(g) Apply a smooth flat skim coat of embedding compound across the entire surface of the TMA.

(h) Place the completed TMA in a labeled Tissue-Tek cassette and store in an air-tight container at −80 °C for up to 3 months.

This step-by-step protocol enables anyone with experience handling tissue to generate TMAs suitable for MALDI-IMS. The storage times indicated above and in subsequent sections enable TMAs to be generated in stages and batch-analyzed. With practice and possibly the assistance of a second person, the workflow should be rapid and efficient. For example, undergraduate students and technicians readily mastered the protocol and currently enable its routine use in the laboratory. However, it is imperative to minimize the time allotted for coring and transferring the fresh-frozen cores to recipient blocks since the sample temperatures should not be permitted to rise above −20 °C for prolonged periods due to potential degradation. The coring and transfer process should take 30 s or less per sample. Complete sample thawing should be avoided. Formalin-fixed tissue cores are less problematic as the sample integrity is preserved by fixation.

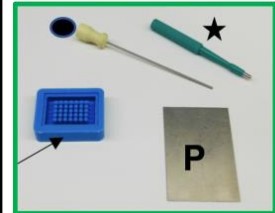
1) Collect materials needed to fabricate the Cryo-TMA: silicone TMA mold (arrow), punch/coring tool (asterisk), stylus (black dot), and metal plate (P).

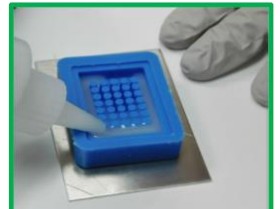
2) Position the silicone mold over the metal plate then fill it with embedding compound. Fill until flush with the arraymold's rim.

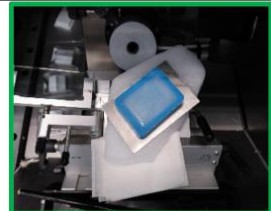
3) Freeze the embedding compound by positioning the filled arraymold with metal plate on top of a flat piece of dry ice. This can be done in a cryostat chamber at -20°C.

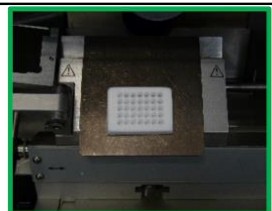
4) Carefully peel away the silicone arraymold to release the newly formed TMA recipient block. Dense arrays with 1mm cores require care to avoid breakage.

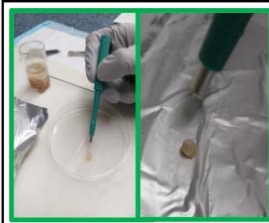
5) Left- Collect fresh, frozen, or formalin-fixed tissue cores using the punch/coring tool. The punch and arraymold diameters should match. Right- Higher magnification of a 3mm brain tissue core.

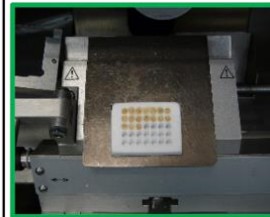
6) Transfer cored samples into the recipient array, one-by-one. Use the stylus to release the tissue from the punch. Perform this step in the cryostat chamber. Use a planned asymmetric map to orient the TMA. Store TMAs at -80°C.

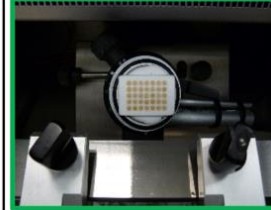
7) For cryo-sectioning, apply a small disc of O.C.T. to the surface of a pre-chilled cryostat chuck and immediately mount and level the completed Cryo-TMA. Orient the TMA block to optimize sectioning.

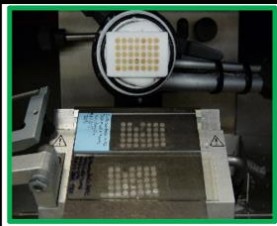
8) Face the block to capture the full array in the cryo-sections. Cut 8-20 μm thick sections. Mount sections onto ITO-coated slides for MALDI-IMS, and adjacent sections onto Plus (+) charged slides for histology.

**Figure 1.** TMA/MALDI-IMS workflow. The major steps used to generate Cryo-TMAs for MALDI-IMS are depicted in captioned Panels 1–8. (1) Tools needed to generate the Cryo-TMAs include a silicone arraymold (arrow), a metal plate (P), a coring/punch tool (asterisk), and a stylus for pushing cored tissue into the arraymolds (black dot). (2) Fill the arraymold with embedding compound. (3) Freeze the embedding compound on a bed of dry ice inside a cryostat chamber ($-20\ ^\circ$C or lower). (4) Release the newly formed frozen recipient block from the arraymold and store it in an air-tight container at $-80\ ^\circ$C. (5) Core tissue samples with a punch. Equilibrate frozen tissue to $-20\ ^\circ$C in a cryostat chamber prior to sampling. (6) Transfer tissue cores to the recipient array block according to a pre-planned asymmetric map to maintain orientation. Apply a skim coat of embedding compound to secure the tissue cores in place. Store completed TMAs at $-80\ ^\circ$C. (7) Mount the TMA block onto a chuck and equilibrate it to the ambient cryostat chamber's temperature ($-20\ ^\circ$C) prior to sectioning. (8) Trim to fully face the block (TMA) and section (8–20 μm thickness). Mount sections onto ITO-coated slides for MALDI-IMS and at least one additional section onto a Plus (+) charged slide for histology. Store TMA slides at $-80\ ^\circ$C until ready for MALDI-IMS.

*2.4. TMA Sectioning and Slide Preparation for MALDI-IMS*

1. Equilibrate the frozen TMA (stored at $-80\ ^\circ$C) to $-18\ ^\circ$C in a cryostat microtome chamber for 20–30 min prior to sectioning.
2. Mount the block onto a cryostat chuck to optimize sectioning of the full TMA, including all tissue cores.
3. Section the TMA at a thickness between 8 μm and 20 μm using a clean, fresh disposable blade for each TMA.
   a. Generate two to four sets of 4 TMA sections.
   b. Use the first 3 adjacent sections/set for MALDI-IMS and the fourth for histologic staining.

    c.    Label and number each slide in the order of sectioning. Label with a pencil or permanent marking pen.

4.    For MALDI-IMS, thaw-mount the cryosections onto indium tin oxide (ITO)-coated slides (Delta Technologies, Loveland, CO, USA).

5.    Desiccate the slides designated for MALDI-IMS at room temperature in a sealed chamber, and then, either store them in an air-tight container at −80 °C for up to 3 months or immediately proceed with sample sublimation.

6.    For histology co-registration with images acquired through MALDI-IMS, thaw-mount cryosections onto Plus-charged glass slides (Thermo Fisher Scientific, Plainville, MA, USA) and air-dry. Either store in an air-tight container (up to 3 months) or immediately proceed with staining protocol below.

    a.    Fix tissue sections in 10% neutral buffered formalin.

    b.    Rinse several times (10–15 dips) in distilled water.

    c.    Stain with Gil's Hematoxylin (Thermo Fisher Scientific, Plainville, MA, USA) according to the manufacturer's instructions.

    d.    Dehydrate tissue sections in graded ethanol solutions (50%, 70%, 95%, 95%, 100%, 100%) for 30 s each, clear in two changes of xylenes (30 s each), and then, preserve under coverglass with Per Mount mounting medium (MilliporeSigma, Burlington, MA, USA) or a comparable product. Store the stained slides at room temperature in a dust-free slide box/holder.

    e.    Scan the slides to generate 3600 DPI resolution images (Epson's Perfection V850 Scanner, Los Alamitos CA, USA) just prior to MALDI-IMS data acquisition.

### 2.5. Matrix Application

The matrix application methods for MALDI-IMS lipidomics with cryosections have been described in detail elsewhere [21,38,39]. Since the goal of this manuscript is to provide a method for generating TMAs for MALDI-IMS lipidomics, the further steps are summary-listed rather than detailed.

1.    Equilibrate stored frozen TMA slides to room temperature.

2.    Rinse in aqueous buffer, such as 50 mM ammonium formate (pH 6.4), to increase lipid ion signal intensities [49].

3.    Vacuum-dry slides for 30 min to promote tissue adhesion to slides.

4.    Sublime the slides with a suitable matrix using a commercial apparatus, such as that from Chemglass Life Sciences (Vineland, NJ, USA).

5.    Choose a matrix appropriate for negative (NIM) and/or positive ion mode (PIM) imaging.

    a.    NIM imaging is optimum for detecting most phospholipids and sulfatides.

    b.    PIM imaging is most suited for detecting ceramides, sphingomyelin, phosphatidylcholine, and cholesterol.

    c.    Use 2,5-dihydroxybenzoic acid (DHB; Sigma-Aldrich Co, St. Louis, MO, USA), 208 $\mu$g/cm$^2$, as a matrix for NIM or PIM [49–52].

        i.    Alternative matrices for NIM or PIM imaging should be considered to optimize the results. See references [53–56].

6.    After sublimation, add external mass-calibration standards (Peptide Calibration Standard II, Bruker Daltonics, Bremen, Germany) by depositing 1 $\mu$L of a standard peptide mixture with 15 mg/mL $\alpha$-cyano-4-hydroxycinnamic acid (HCCA) as a matrix, as recommended by the manufacturer.

    a.    The mass range is from 377 Da to 2463 Da.

    b.    Enables mass accuracy determinations for phospholipids and sphingolipids.

*2.6. MALDI-IMS*

The methods for performing MALDI-IMS lipidomics have been described and are widely available [21,38,39]. The major steps used to analyze data for this article are summarized below.

1.  TMA slides sublimed with DHB as the matrix were imaged in the negative and positive ion modes using a reflectron geometry MALDI-time-of-flight (TOF)/TOF mass spectrometer (Ultraflextreme, Bruker Daltonics, Bremen, Germany).
2.  Data acquisition (restricted to a mass range of 600–1200 Da) was performed by focusing a Smartbeam II Nd:YAG laser with a spatial resolution of ~100 $\mu m^2$ [50,51,57].
3.  Regions of interest were selected based on co-registration with adjacent hematoxylin-stained slides.
4.  Data sequence preparation, normalization to total ion counts, and visualization were carried out using FlexImaging software (v 4.0, Bruker Daltonics, Bremen, Germany).
5.  Data processing, which included normalization, baseline correction, peak defining, and recalibration, was performed with ClinProTools v3.0 (Bruker Daltonics, Bremen, Germany).
6.  Signals corresponding to specific $m/z$ values were visualized using pseudo-colored intensities.
7.  Statistical analysis, including Principal Component Analysis (PCA), was carried out using ClinProTools v3.0 (Bruker Daltonics, Bremen, Germany).
8.  NIM lipid identification was accomplished through a comparison of precursor and product ion $m/z$ values with corresponding data in the LIPID MAPS prediction tools database (https://www.lipidmaps.org/tools/structuredrawing/GP_p_form. php, accessed on 8 December 2022).
    a.  Using tandem mass spectrometry (MS/MS), analytes were fragmented, and their product ions were collected in the MS/MS spectra.
    b.  The parent ion and all fragments were used to search the LIPID MAPS database and assign structure/identity.
    c.  However, for low-intensity lipid ions, the MS/MS spectra were not informative, necessitating tentative assignments made by matching our $m/z$ values with published data [54,55,58].
9.  PIM lipid identification was more challenging due to the presence of multiple adducts ($H^+$, $Na^+$, $K^+$, etc.), resulting in complex spectra. Structural identification was often difficult with TOF since the same ion could appear as multiple adducts. For these studies, we did not perform MS/MS to definitively identify ambiguous lipids detected in the PIM. Instead, tentative assignments were made using the literature where the same $m/z$ was detected under similar conditions [52–55,59–63].
10. The Rapid Peak Alignment Method (RPAM) [64] was used to simultaneously process the MALDI data across the TMA.
    a.  In brief, RPAM replaces manual peak alignments and reduces the data processing time for 24 samples from 10 or more hours to approximately 90 min [64].
    b.  RPAM data can be transferred to statistical packages for analysis.
    c.  The RPAM algorithm greatly facilitates intra- and inter-group comparisons of lipid ion expression and abundance [64].

*2.7. Statistical Analyses and Graphics*

1.  Data were exported to Excel for re-organization.
2.  Excel was used to generate data bar plots for illustrating inter-group percentage differences in lipid expression.
3.  Graphpad Prism 9 (San Diego, CA, USA) and Number Cruncher Statistical Systems (NCSS) (Kaysville, UT, USA) software were used to generate graphs and analyze data with Student T and Wilcoxon Signed Rank tests. Results were corrected for a 5% false discovery rate (FDR).

## 3. Results

Experimental data were used to illustrate the utility of TMAs for MALDI-IMS lipidomics. Critical comments include the effects of different embedding compounds, fresh-frozen versus formalin-fixed tissue, and feasibility for the analysis of disease effects.

### 3.1. Embedding Compound Qualitative Differences

1.  mOCT produced very low background signals and exhibited ample firmness for reproducible cryostat sectioning.
2.  The commercial OCT embedding compound was unsuitable for MALDI-IMS lipidomics due to high background signals. Although this limitation was known [46], commercial OCT was tested to provide evidence of its unsuitability for lipidomics and definitively discourage its use despite ready availability.
3.  Two percent CMC produced very low background noise but proved too soft to generate consistent replicate and flat cryo-sections of brain TMAs.
    a.  We also tested 4% CMC, which produced very low background signals but was unsuitable for TMAs due to extreme brittleness, particularly for brain tissue sectioning.

Conclusions

Select an embedding compound that optimizes the signal-to-noise ratio for specific tissues.

Select embedding compound preparations that reproducibly generate high-quality TMA cryosections.

### 3.2. Effects of Embedding Compound on Disease Characterization with Fresh Frozen Tissue TMAs (Figure 2)

1.  Paired control and ethanol rat frontal lobe WM cores (3 mm) were embedded in mOCT and 2% CMC (hybrid recipient block—see Figure 2 legend).
2.  Data analysis focused on sulfatide ST(42:2), $m/z$ 888.772, which was identified as previously described [21,38,39,50]. A mixture of calibration standards with $m/z$ values spanning the range of analytes of interest and applied to the MALDI target was used to visualize the tissue distribution and relative intensity of each ion at every pixel using a pseudo-color scale. In the NIM, pseudo-colored images demonstrated the following:
    a.  Brighter (more intense) MALDI-IMS signals for cores of frontal lobe WM from the same animals embedded in 2% CMC compared to those with mOCT.
    b.  The ethanol exposure-associated higher signal intensities were more conspicuous for samples embedded in 2% CMC compared to those with mOCT.
    c.  However, the 2% CMC embedding compound posed challenges for generating multiple replicate TMA sections.

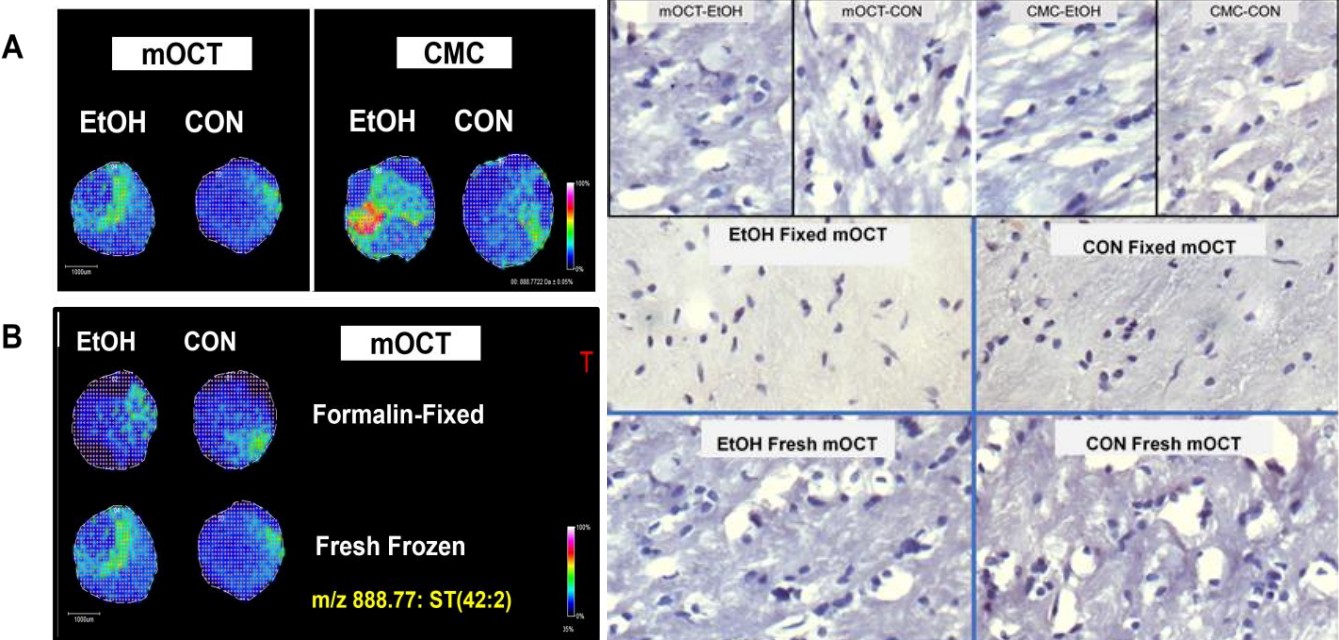

**Figure 2.** Example images illustrating the impact of the embedding compound and tissue preparation on the characterization of altered ST(42:2) expression following chronic ethanol exposure. ST(42:2) is a sulfatide with an *m/z* of 888.8. Frontal lobe WM cores (3 mm diameter) from adult Long Evans rats maintained for 6 weeks on isocaloric liquid diets that contained 36% (EtOH) or 0% (CON) caloric ethanol were used to generate TMAs in hybrid recipient blocks constructed with mOCT in one half and 2% CMC in the other half. To achieve this, two glass slides were stood together in the center of the arraymold, aligned with its shorter axis as the embedding compounds solidified on either side of the dam. Cryosections (8 μm-thick) of the TMAs were sublimated with DHB and imaged via MALDI-IMS in the negative ion mode (NIM). Signal intensities were pseudo-colored based on an internal standard reference scale. (**A**) Comparisons between mOCT and 2% CMC in fresh-frozen tissue. (**B**) Comparisons between formalin-fixed and fresh-frozen EtOH and CON samples embedded in mOCT. The panel to the right shows hematoxylin-stained histological sections of white matter in the cores imaged on the left. The small round-to-oval blue dot-like structures are nuclei of glial cells. The large white spaces associated with fresh tissue sections correspond to freeze artifacts common to white matter, regardless of the embedding compound. Note the absence of similar artifacts in fixed tissue (original magnification, ×400 for all histology images).

Conclusions

Interpret results obtained with samples embedded in the same compound and analyzed simultaneously under identical conditions.

Use an embedding compound that enables the optimum detection of lipids with a broad range of *m/z* values (corresponding to the calibration standards) in both control and experimental/diseased samples.

Consider technical feasibility challenges in planning experiments.

*3.3. Fresh Frozen vs. Formalin-Fixed Tissue TMAs*

1. NIM MALDI-IMS was performed on paired fresh-frozen and formalin-fixed frontal lobe WM cores embedded in an mOCT arraymold recipient block.

   a. By slicing the brains in the coronal plane at the temporal tips, 3 mm cores of central white matter were obtained. The small amounts of peripheral contamination from the adjacent cortex were easy to exclude from the analysis by marking the co-registered images.

2.  Signal intensities were similar for the same samples and tended to be higher (brighter) for fresh-frozen compared than for formalin-fixed tissue cores embedded in mOCT (Figure 2B).

3.  Example graphed results from individual paired samples analyzed via NIM MALDI-IMS depict peak profiles corresponding to the 21 lipids detected in the fresh-frozen and fixed samples (Figure 3).

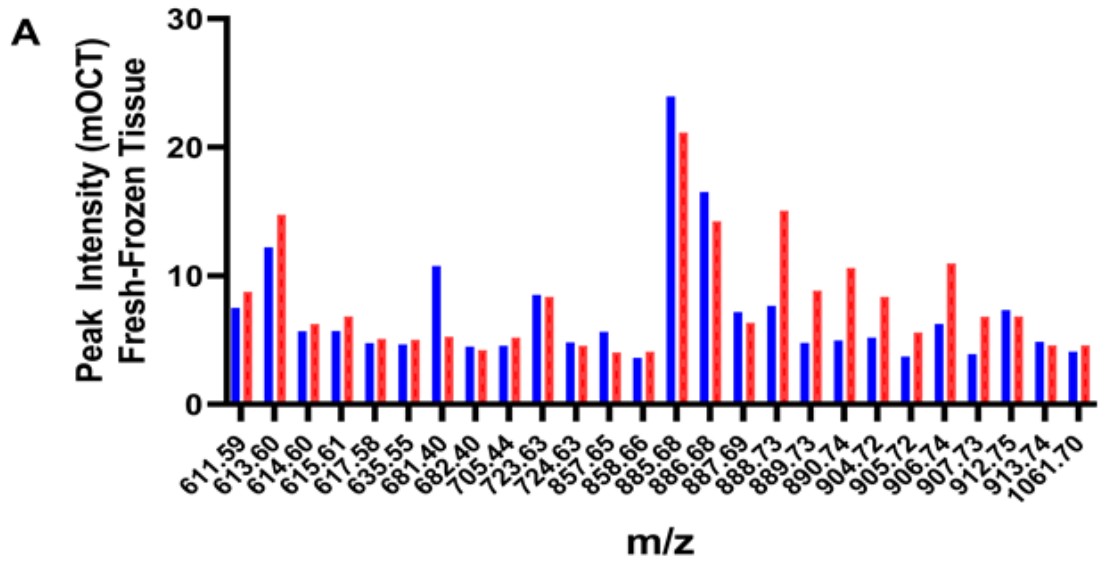

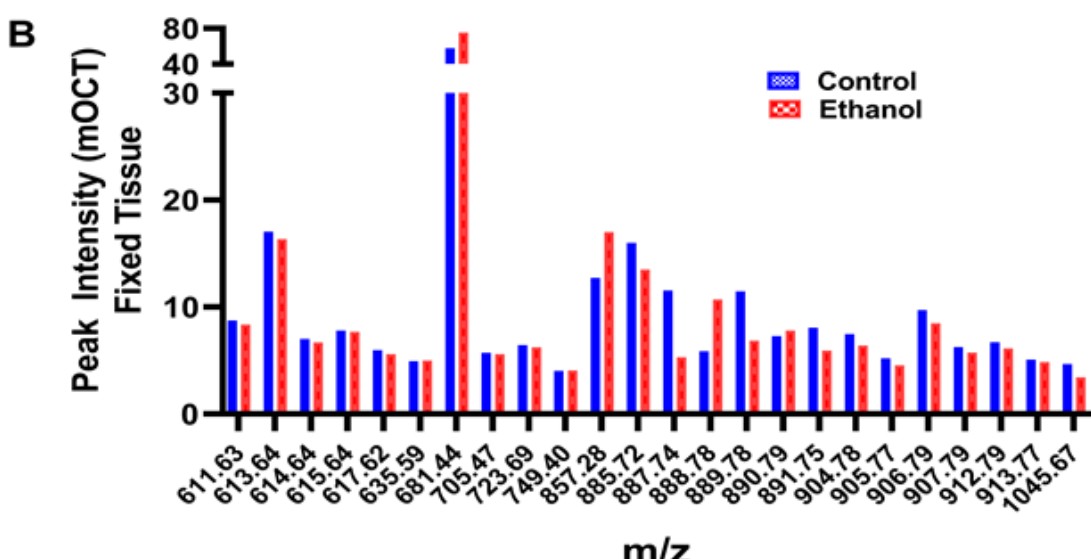

**Figure 3.** Peak intensity spectra corresponding to lipids detected in mOCT-embedded fresh-frozen and formalin-fixed frontal WM from control or ethanol-exposed rats. The TMAs were generated with paired tissue cores (3 mm diameter) from the same brains (*n* = 4 rats/group) (see Methods). Data were acquired via NIM MALDI-IMS. Unique peaks were excluded from these comparisons. Note the extensive overlap of signal intensities within a group, i.e., (**A**) frozen or (**B**) fixed, and conspicuous peak profile distinctions based on tissue preservation (Panel (**A**) vs. Panel (**B**)).

4.  Results were nearly identical for control and ethanol brains that were either fresh-frozen (Figure 3A) or formalin-fixed (Figure 3B).

5.  However, the signal intensities (peak heights) corresponding to individual lipids differed for formalin fixed versus fresh-frozen tissue samples (Figure 3A,B).

Conclusions

TMAs generated with formalin-fixed tissue are suitable for MALDI-IMS lipidomics.

The finding that formalin-fixed brain tissue could be used for MALDI-IMS is consistent with earlier reports [33,34,65].

Due to non-identical peak profiles with the same embedding compound, as previously reported [65], formalin-fixed and fresh-frozen sample results should not be compared.

The suitability of formalin-fixed tissue for TMA MALDI-IMS lipidomics expands opportunities for large-scale retrospective studies, including the analysis of human diseases.

Although formalin fixation with paraffin embedding is suitable for metabolomics [66], it markedly distorts lipidomics [60]. However, formalin fixation alone with proper and simultaneously analyzed controls enables reliable inter-group comparisons. Correspondingly, we detected nearly identical lipid expression profiles but differences in peak intensities (lipid abundances) in ethanol-exposed versus control samples (see Figure 3). We strongly discourage the use of paraffin-embedded samples due to the unpredictable and often substantial removal of lipids in the dehydration and clearing steps, rendering the samples unsuitable for MALDI-IMS lipidomics [60].

Formalin fixation generally does not interfere with lipidomic profiling through MALDI [67]. However, prolonged formalin fixation can adversely affect the structural integrity of lipids that contain primary amines via cross-linking [60]. Therefore, formalin fixation should be optimized by the following: (1) keeping the specimen thickness to 3 mm or less; (2) using neutral-buffered formalin; (3) limiting fixation times to 48 or 72 h; (4) storing postfixed tissue in PBS containing 0.01% sodium azide at 4 °C up to a maximum of 3 weeks.

DHB is generally used for positive- and negative-mode imaging [54,55], but some accounts have reported lower lipid detection sensitivity with DHB compared to that with other matrices for negative-mode imaging [68]. However, after experimenting with different matrices, we found that DHB provides broad lipid coverage with many sphingolipids and phospholipids detected in positive and negative ion modes within the mass range of interest, i.e., $m/z$ = 600–1200 [21,33,50].

### 3.4. Embedding Compound Effects on Lipid Peak Profiles (Figure 4)

1. NIM MALDI-IMS detected 140 shared lipid peaks ($m/z$ 600–1200) in TMAs generated with control and ethanol-exposed fresh-frozen or formalin-fixed rat frontal WM cores embedded in mOCT or 2% CMC ($n$ = 4 rats/group).
2. Principal Component Analysis (PCA) plots generated with ClinProTools, v3 demonstrated clear separation of control and ethanol fresh frozen versus formalin-fixed samples embedded in mOCT (Figure 4A) or CMC (Figure 4B).
3. PCA plots demonstrated the effects of ethanol, which were better distinguished with the 2% CMC embedding compound than with mOCT.
4. These studies demonstrate the effects of the treatment/disease model but the more prominent effects of tissue processing and the embedding compound.
5. Data generated with cores sampled from the same brains but spatially positioned in different regions of the TMA had a less than 5% mean coefficient of variation.

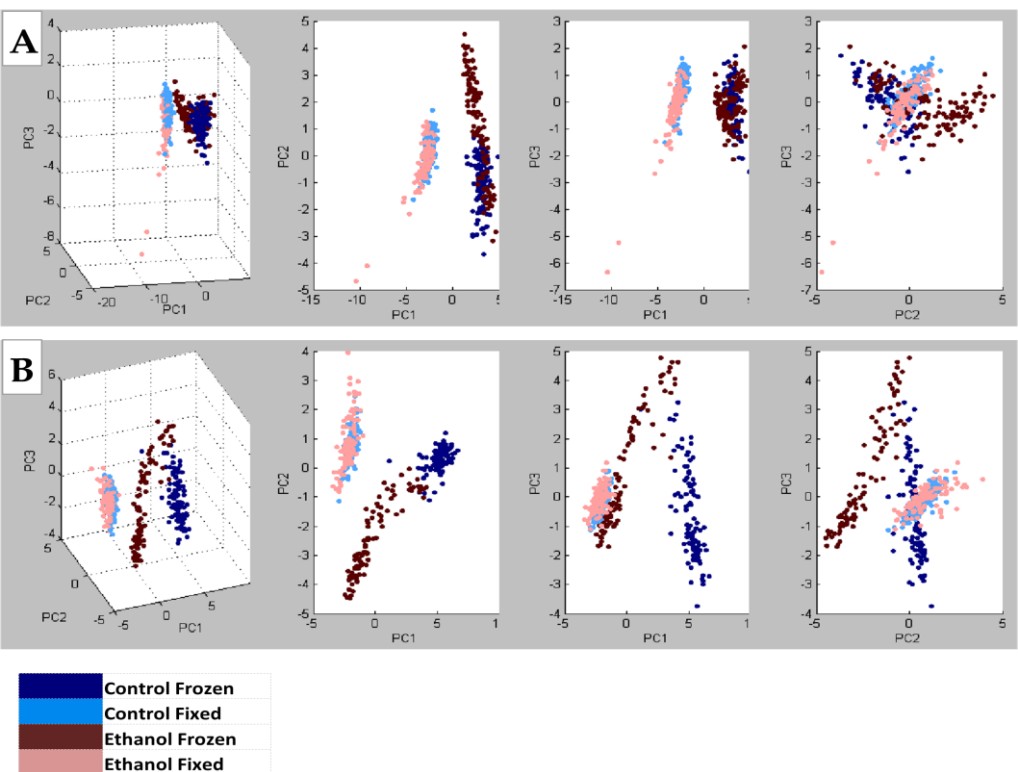

**Figure 4.** Principal Component Analysis (PCA) plots corresponding to the full spectra of lipids detected in fresh-frozen or formalin-fixed frontal lobe samples from control and ethanol-exposed rats (See Supplementary Figure S1). TMAs generated with (**A**) mOCT or (**B**) 2% CMC as the embedding medium were analyzed via MALDI-IMS with NIM signal acquisitions. PCA plots were generated with ClinProTools, v3. Note the clear separation of control and ethanol fresh-frozen versus formalin-fixed samples embedded in mOCT or CMC. However, with data combined, the overlap of signals was more tightly related to tissue processing (freezing of fixation) than experimental treatment (ethanol or control diet) with either mOCT or CMC as the embedding compound.

Conclusions

TMAs are suitable for characterizing normal and disease-related lipid profiles in tissue.

Since formalin-fixed tissue is suitable for detailed MALDI-IMS lipidomics, the approach could be used to analyze pathological shifts in lipid expression and responses to treatment in large-scale experimental and human disease states.

Tissue processing and selection of embedding compounds must be uniform for reliable data analysis via TMA-MALDI-IMS.

### 3.5. Illustrated Application of MALDI-IMS TMA Methodology

1.  Chronic + Binge Ethanol Exposure model [69]: Adult Long Evans rats were maintained for 8 weeks on isocaloric Lieber-DeCarli liquid diets containing 24% or 0% caloric ethanol liquid diets (BioServ, Frenchtown, NJ, USA), and during the last 3 weeks, they were gavage-binged with 2 g/kg ethanol or saline in the liquid diet (2.5 mL total volume) on Tuesdays, Thursdays, and Saturdays.
2.  Fresh-frozen WM cores (2 mm) from six rats per group were used to generate mOCT-embedded TMAs.
3.  The NIM and PIM frontal white matter MALDI-IMS lipidomics spectra were distinct (Figure 5). The control and ethanol samples mainly differed with respect to selected peak intensities, rather than specific lipids detected within the NIM or PIM.

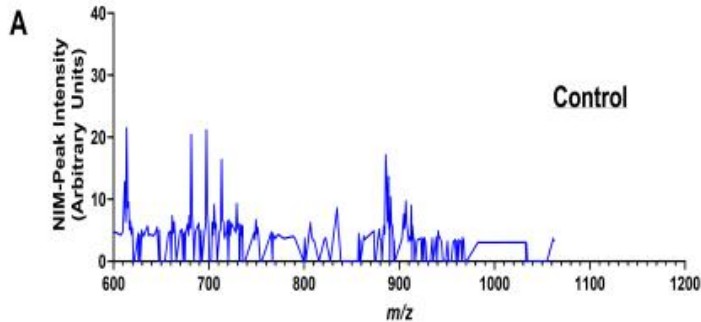

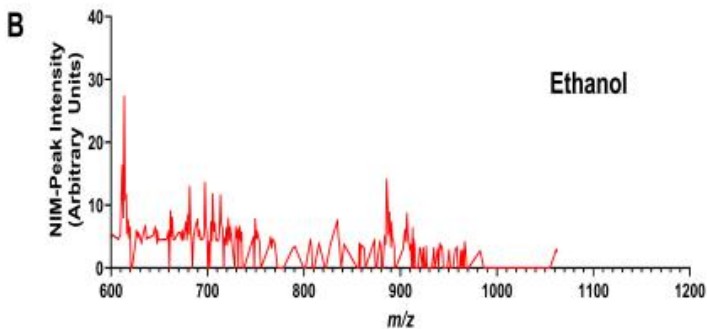

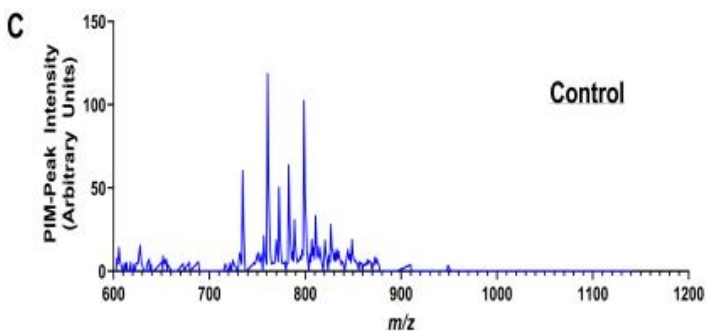

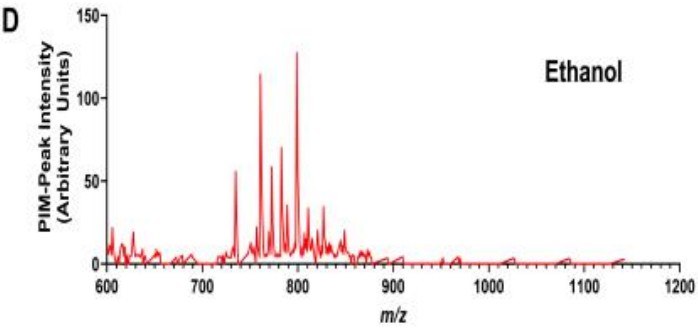

**Figure 5.** MALDI-IMS NIM and PIM spectra from a TMA constructed with fresh-frozen, 2 mm cores of frontal lobe white matter embedded in mOCT. Spectra correspond to relative intensities (abundances) of lipid ions between $m/z$ 600 and 1200 detected by MALDI (**A,B**) NIM and (**C,D**) PIM. Data were acquired by rasterizing across the TMA WM cores of (**A,C**) control and (**B,D**) chronic ethanol-fed rats. The peak intensities (arbitrary units) reflect averaged results from four rats per group.

4. Further MALDI-IMS analysis was focused on phosphatidylinositol (PI) data (Figure 6A) acquired in NIM and phosphatidylcholine (PC) data (Figure 6B) acquired in the PIM

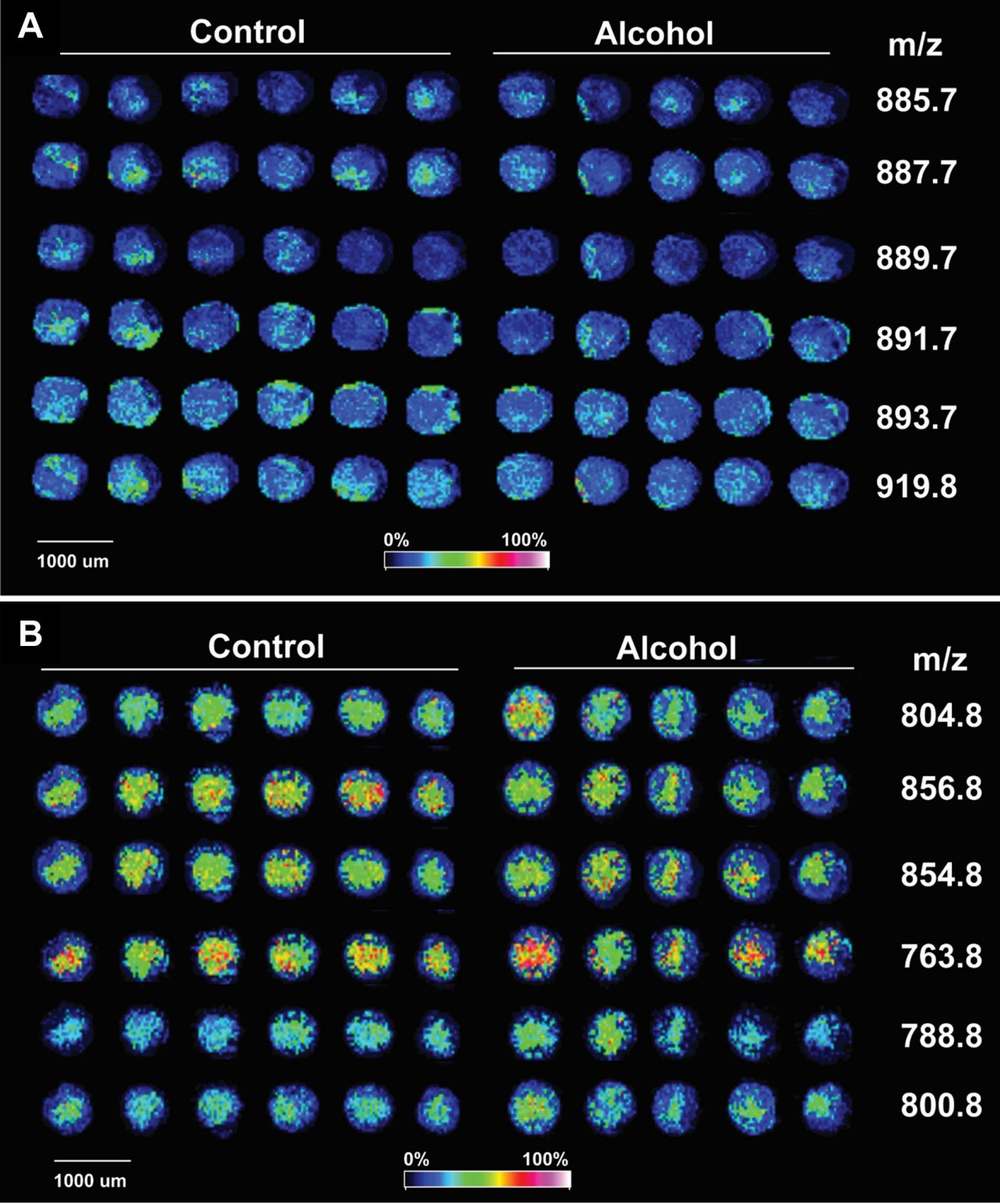

**Figure 6.** MALDI-IMS TMAs of frontal lobe white matter. MALDI-IMS TMAs generated with fresh frozen frontal lobe white matter cores from control and ethanol-fed rats were embedded in mOCT recipient blocks and imaged in the (**A**) negative or (**B**) positive ion mode. Lipid ion signals were pseudo-colored to reflect signal intensities. (**A**) Example NIM results correspond to phosphatidyli­nositols, and the (**B**) PIM results correspond to phosphatidylcholines. See Supplementary Figure S2 for representative histologic images.

1. Table 1 lists the 10 PIs and 11 PCs identified via tandem mass spectrometry (MS/MS) coupled with LIPID MAPS database searches and published reports [39].

**Table 1.** Phosphatidylinositols and phosphatidylcholines detected in frontal lobe white matter via MALDI-IMS.

| Code | Phosphatidylinositols-NIM | Ionization Form | *m/z* |
|---|---|---|---|
| a | PI(34:1) | [M⁻H]⁻ | 835.7 |
| b | PI(36:4) | [M⁻H]⁻ | 857.6 |
| c | PI(O-38:3)/PI(P-38:2) | [M⁻H]⁻ | 873.2 |
| d | PI(38:5) | [M⁻H]⁻ | 883.7 |
| e | PI(38:4) | [M⁻H]⁻ | 885.7 |
| f | PI(38:3) | [M⁻H]⁻ | 887.7 |
| g | PI(38:2) | [M⁻H]⁻ | 889.7 |
| h | PI(38:1) | [M⁻H]⁻ | 891.7 |
| i | PI(38:0) | [M⁻H]⁻ | 893.7 |
| j | PI(40:1)/PI(P-41:0)/LPIM2(18:2) | [M⁻H]⁻ | 919.8 |
| | **Phosphatidylcholines-PIM** | **Ionization Form** | *m/z* |
| a | (1) PC(36:1); (2) pPC(36:4); (3) PC(34:6) | (1) [M⁺H]⁺; (2) [M⁺Na]⁺; (3) [M⁺K]⁺ | 788.8 |
| b | (1) PC(36:7); (2) PC(O-36:0); (3) PC(34:1) | (1) [M⁺Na]⁺; (2) [M⁺Na]⁺; (3) [M⁺K]⁺ | 798.8 |
| c | (1) PC(38:7); (2) PC(36:4) | (1) [M+H]⁺; (2) [M+Na]⁺ | 804.8 |
| d | (1) PC(40:9); (2) PC(O-40:2); (3) PC(38:6); 4) PC(36:0) | (1) [M⁺H]⁺; (2) [M⁺H]⁺; (3) [M⁺Na]⁺; (4) [M⁺K]⁺ | 828.8 |
| e | (1) PC(42:10); (2) PC(40:7); (3) PC(38:1) | (1) [M⁺H]⁺; (2) [M⁺Na]⁺; (3) [M⁺K]⁺ | 854.8 |
| f | (1) PC(42:9); (2) PC(40:6) | (1) [M⁺H]⁺; (2) [M⁺Na]⁺ | 856.8 |
| g | (1) PC(O-38:2); (2) PC(36:6); (3) PC(34:0) | (1) [M⁺H]⁺; (2) [M⁺Na]⁺; (3) [M+K]⁺ | 800.8 |
| h | PC(32:0) | [M⁺H]⁺ | 734.8 |
| i | PC(34:0) | [M⁺H]⁺ | 762.8 |
| j | PC(34:1) | [M⁺H]⁺ | 760.8 |
| k | PC(36:3) | [M⁺K⁻N(CH₃)₃]⁺ | 763.8 |

Phosphatidylinositols and phosphatidylcholines detected in frontal lobe white matter via MALDI-IMS. Cores (2 mm) of fresh-frozen frontal white matter from control and ethanol-fed Long Evans rats were embedded in mOCT (*n* = 6 samples/group). The TMAs, sublimated with DHB, were imaged in the negative ion mode (NIM) and positive ion mode (PIM). The 10 PI and 11 PC lipid ions detected in all samples are listed with their ionization forms and *m/z* values. Codes (first column) correspond to graphed results displayed in Figure 7.

2. Inter-group differences in the mean peak intensity for each lipid were assessed statistically with Student *t*-tests. The calculated percentage differences in the ethanol relative to control samples are depicted in databar plots alongside the *p*-values for the individual *t*-tests (Figure 7A,C).

3. However, the overall effects of ethanol on PI (Figure 7B) and PC (Figure 7D) expression were further assessed using the Wilcoxon Signed Rank Test to determine if the median percentage differences differed significantly from 0.00. The test demonstrated overall significantly reduced PI ($p = 0.002$) and increased PC ($p = 0.003$) expression in ethanol-exposed frontal WM.

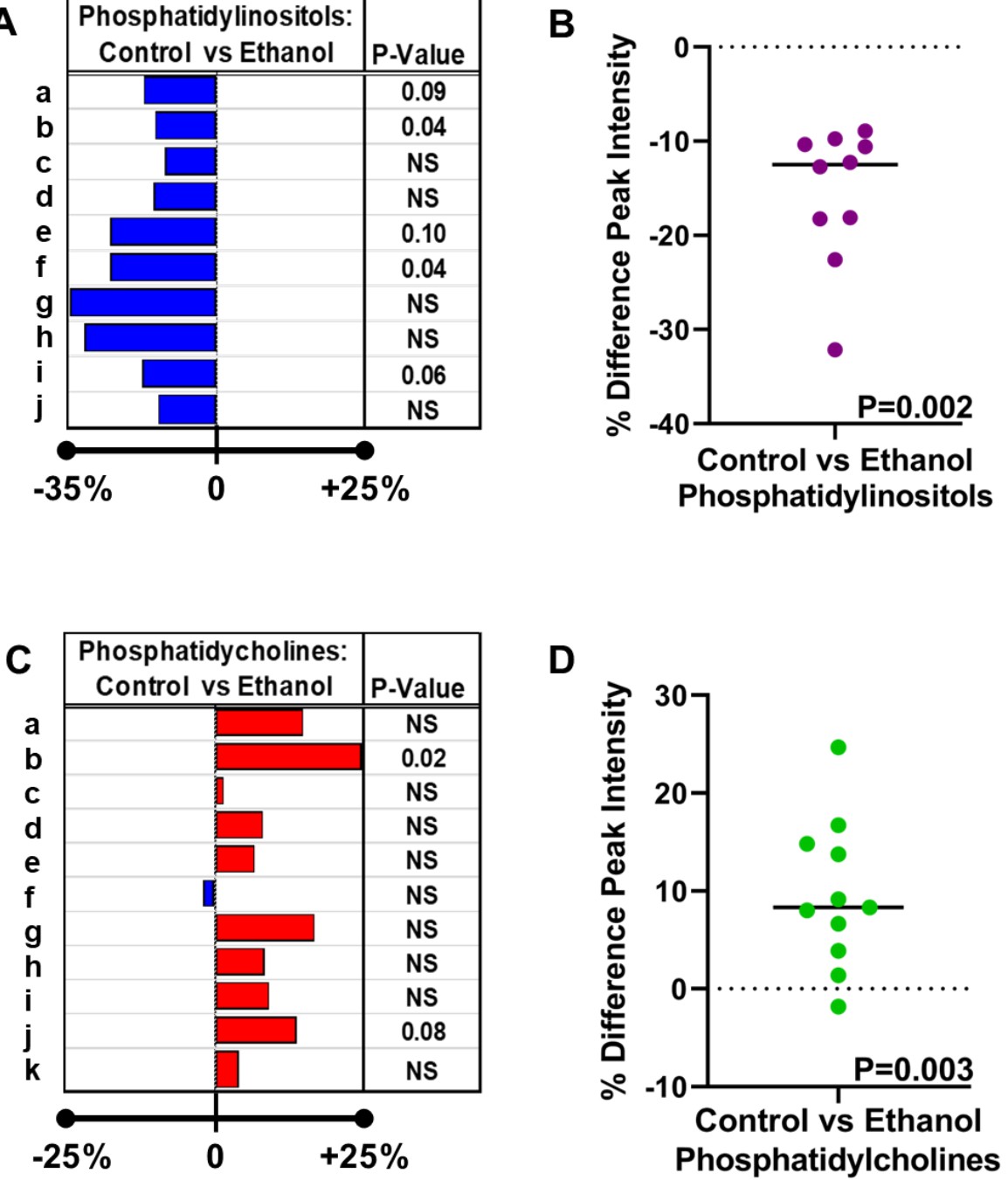

**Figure 7.** Databar plots were generated to reflect percentage differences in frontal lobe white matter levels of (**A**) phosphatidylinositol (PI) and (**C**) phosphatidylcholine (PC) expression resulting from chronic dietary ethanol exposure. This study utilized a chronic+binge ethanol feeding model with six rats per group. A TMA generated with fresh-frozen tissue embedded in mOCT was imaged. Ten PIs were detected in the NIM and 11 PCs in the PIM (see Table 1). The calculated paired percentage differences in lipid expression were used for the Databar plots, such that ethanol-associated reductions are represented by blue bars to the left and increases by red bars to the right. Inter-group differences were assessed by performing Student *t*-tests with a 5% false discovery rate. Significant differences ($p < 0.05$) and trend effects ($0.05 < p < 0.10$) were determined using Prism Graphpad v9.4. NS = not statistically significant. (**B**,**D**) Scatter plots displaying the distribution of differences in (**B**) PI and (**D**) PC expression between ethanol and control frontal lobe white matter samples. The horizontal bar corresponds to the median. The hypothesis tested was that the inter-group differences equaled 0. The Wilcoxon Signed Rank test was statistically significant for the PI and PC grouped results.

Conclusions

TMAs are suitable for characterizing disease-related alterations in brain WM lipid expression.

Additional studies showed that data generated with cores sampled from the same brains but spatially positioned in different regions of the TMA had a less than 5% mean coefficient of variation.

## 4. Discussion

- This work provides a practical, high throughput approach for generating TMAs that are suitable for MALDI-IMS lipidomic studies.
- Although the efforts were focused on brain white matter biochemical histology, the methods can be successfully applied to a broad range of tissues.
- The demonstrated feasibility of using formalin-fixed tissue expands opportunities to conduct large-scale MALDI-IMS lipidomic studies of archival specimens to study the effects of disease in humans and experimental models.
- Although the TMA approach necessitates the inclusion of smaller samples for analysis compared to that ordinarily used for MALDI-IMS and therefore bears the risk of missing data, the streamlined higher throughput strategy enabling uniform sample handling and processing and simultaneous imaging of replicate, positive, and negative control samples under the same conditions outweigh potential limitations related to smaller sample sizes. Moreover, the TMA approach bolsters overall scientific rigor.
- Potential limitations posed by the need to analyze multi-sample lipid profile TMA datasets can be resolved using the open-access rapid peak alignment method (RPAM) [64].
- The use of TMAs for MALDI-IMS could potentially facilitate the better characterization of white matter myelin-associated pathologies that correlate with disease progression or responses to treatment, particularly with respect to neurodegeneration.

**Supplementary Materials:** The following supporting information can be downloaded at: https://www.mdpi.com/article/10.3390/applbiosci2020013/s1, Figure S1: MALDI-IMS spectra corresponding to (A,B,E,F) fresh frozen or (C,D,G,H) formalin-fixed frontal lobe white matter samples from (A,C,E,G) control and (B,D,F,H) ethanol-fed rats. 3-mm tissue cores were embedded in a hybrid (A–D) mOCT/(E–H) 2% CMC TMA and following DHB sublimation coating, the samples were imaged in the negative ion mode (NIM) along with calibration standards used to assess the relative abundance (peak intensity-arbitrary units) of each lipid ion. Each spectrum represents averaged results from 2 rats per group. See Figure S2; Figure S2: Example optical images of Hematoxylin-stained TMA frontal white matter cores from control and chronic ethanol-fed rats corresponding to the 4th core from the left in each group shown in Figure 6. Hematoxylin labels nuclei. The cores were photographed at different magnifications to reveal the glial-predominant and microvascular composition of white matter. Example oligodendrocytes with dot-like nuclei and astrocytes with oval nuclei are depicted to the left of the O's or A's. Microvessels with narrow lumens are marked with arrows. The red squares show the regions of higher magnification depicted in the immediately below panels.

**Author Contributions:** All authors contributed to the research efforts required to generate this manuscript and reviewed this manuscript. I.G.-R., L.N. and M.T. contributed equally to method development. E.B.Y. supervised the MALDI and contributed primary data. S.M.d.l.M. conceived the idea, contributed to the research design, performed data analysis, and wrote and revised the manuscript. All authors have read and agreed to the published version of the manuscript.

**Funding:** This research was funded by grants from the National Institutes of Health-National Institute of Alcohol Abuse and Alcoholism: AA-011431 and AA-024018.

**Institutional Review Board Statement:** The animal study protocol was approved by the Lifespan Institutional Animal Care and Use Committee (IACUC) of Rhode Island Hospital (CMTT000615, initially approved 11 February 2015) for studies involving animals.

**Informed Consent Statement:** Not applicable.

**Data Availability Statement:** Not applicable.

**Acknowledgments:** The authors acknowledge William Pelit, Department of Chemistry, Brown University, for assisting with the retrieval of MALDI spectra from data files.

**Conflicts of Interest:** The authors declare no conflict of interest.

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
