# Peer review of "Tissue Microarray Lipidomic Imaging Mass Spectrometry Method: Application to the Study of Alcohol-Related White Matter Neurodegeneration"

_2813-0464, doi:10.3390/applbiosci2020013_

Round 1

Reviewer 1 Report

The authors provide a practical, high-throughput approach for generating TMAs suitable for MALDI-IMS lipidomic studies. The execution of this study is perfect and the results are highly interesting and usable for MALDI-IMS users.

Please make interpunctuation at titles and in the description of the method congruent.  

E. g. 2.1. Overview. and line 158

Compared with other published material,TMA is an important tool for histologically examining a large number of samples in parallel. This is especially true for MALDI IMS as an imaging technique, as lipids and proteins can also be analyzed without labeling.

The resolution has to be improved when published.

Author Response

The authors appreciate the very thoughtful and thorough review. We have made the corrections as suggested.

We have revised the interpunction at the titles under Methods.

We have provided higher-quality images of the MALDI-IMS figures for publication. The DPI is at least 300.

Reviewer 2 Report

Isabel et al. showed the application of a sample preparation method, namely, tissue microarray (TMA), to study lipids in an experimental rat model of chronic alcohol feeding by MALDI-IMS. 

I recommend accepting the manuscript. Before publication, authors are asked to address the following points:

 1.     Besides TMA, authors could prepare some samples by conventional protocol and analyze both by MALDI-IMS to see the consistency of the findings.

2.     Line 385. Authors claimed that formalin-fixed tissues are suitable for lipidomics. On the other hand, it is reported that formalin fixation causes the oxidation of endogenous lipids and the formation of products between lipids and fixative ingredients (https://doi.org/10.3390/ph15111307). How would authors convince the readers?

3.     The described protocol requires a long time for sample preparation. Also, the sample exposed to several different conditions (e.g., -80°C, -20°C, semi-thaw, etc.) during the procedure. It seems hard to maintain the quality of the samples. Authors should discuss it.

4.     Line 119. Why was OCT considered for embedding? It is well-known that OCT reduces the spectra quality significantly.

5.     DHB matrix is typically used for positive ion mode. But the authors used it for both positive and negative modes (line 281). Why?

6.     Line 349. How did the authors confirm m/z 888.772 as sulfatide ST (42:2)?

7.     Figure 2&5. An optical and/or histological image(s) should be provided.

8.     Figure 1. The resolution and the size of the images should be improved.

9.     TMA has already been described in some previous literature. How is the current protocol different from those reported previously?

10. It would be better to include the major findings (regarding lipidomics in model mice) in the abstract.

11. Representative mass spectra should be provided.

12. Figure 4. Visibility should be increased.

13. Line 268. Remove the redundancy and correct the spelling regarding the DHB.

14. “m/z” should be italicized.

Author Response

The authors very much appreciate the thorough and thoughtful review. The suggestions made will improve the manuscript's quality. Thank you. We've made all of the recommended changes.

  1. Besides TMA, authors could prepare some samples by conventional protocol and analyze both by MALDI-IMS to see the consistency of the findings.

We have published several manuscripts using MALDI lipidomics without the benefit of TMAs. We now comment on this earlier approach. However, our interest in analyzing multiple samples for grouped analysis requires simultaneous sample processing and analysis. Due to concerns about handling individual samples over time by potentially different technologists, it became clear that the only scientifically robust approach would be to generate TMAs.  Therefore, we started from scratch and systematically designed a protocol suitable for TMA MALDI-IMS lipidomics. We have added a comment that reflects these concepts in the Introduction, last sentence of paragraph d.

  1. Line 385. Authors claimed that formalin-fixed tissues are suitable for lipidomics. On the other hand, it is reported that formalin fixation causes the oxidation of endogenous lipids and the formation of products between lipids and fixative ingredients (https://doi.org/10.3390/ph15111307). How would authors convince the readers?

At the end of Section 3.3 (Conclusions), we note that the main concern pertains to the use of paraffin-embedded samples-which should not be used. With proper controls, formalin fixation artifacts (without paraffin-embedding) are similar across all samples, enabling inter-group comparisons. The abovementioned reference is now cited in relation to this concern.

  1. The described protocol requires a long time for sample preparation. Also, the sample was exposed to several different conditions (e.g., -80°C, -20°C, semi-thaw, etc.) during the procedure. It seems hard to maintain the quality of the samples. Authors should discuss it.

We have included a comment at the end of Section 2.3 to clarify the efficiency of sample processing with the protocol provided in this manuscript (Section 2.3.h).

  1. Line 119. Why was OCT considered for embedding? It is well-known that OCT reduces the spectra quality significantly.

An explanation for testing and not recommending OCT for MALDI lipidomics has been added under Results 3.1.2.

  1. DHB matrix is typically used for positive ion mode. But the authors used it for both positive and negative modes (line 281). Why?

We have successfully used DHB for both positive and negative modes. Using DHB for both NIM and PIM, 1) we have had success characterizing disease models and human diseases and 2) with immediately adjacent sections of the same sample preparations, we have been able to extract data pertaining to a broad range of lipids (m/z’s) expressed.  The lower signal intensities associated with DHB compared with other matrices for NIM imaging have not hampered our ability to distinguish the effects of diseases or exposures. We have added comments along these lines including citations that support these claims under “Conclusions” after Section 3.3.

  1. Line 349. How did the authors confirm m/z888.772 as sulfatide ST (42:2)?

We have added citations that include our detailed methods for lipid identification. In addition, we have provided our methodological approach for lipid identification in the MALDI-IMS sections 2.6.8 and 2.6.9.

7. Figure 2&5. An optical and/or histological image(s) should be provided.

Representative histological images are provided in Figure 2 and for Figure 5 (now Figure 6), we have added Supplementary Figure 2.

  1. Figure 1. The resolution and the size of the images should be improved.

Figure 1 has been enlarged. The images are clear after enlargement.

  1. TMA has already been described in some previous literature. How is the current protocol different from those reported previously?

The TMA methods reported herein focus on the use of TMAs for MALDI lipidomics and illustrate the practical utility of the approach using experimental data. We have included this information in the Introduction, Section 1d.

  1. It would be better to include the major findings (regarding lipidomics in model mice) in the abstract.

We have added information about the experimental model findings in the Abstract.

  1. Representative mass spectra should be provided.

            Representative mass spectra are now included in Figure 5 (new) and Supplementary Figure 2

  1. Figure 4. Visibility should be increased.

The figure has been enlarged and its resolution increased to 300 dpi

  1. Line 268. Remove the redundancy and correct the spelling regarding the DHB.

Dihydroxybenzoic acid spelling was corrected and the redundancy in the phrase has been removed.

  1. m/z” should be italicized.

            Done

Reviewer 3 Report

This paper will be of considerable interest to bioscientists. It is clearly written. I do have a few questions about compound identification but these can be easily delt with by the authors.

Comments

1. The term "air tight container" was used in many places, what does this actually mean?

2. Line 300. The link does not work. Do the authors mean this link https://www.lipidmaps.org/tools/structuredrawing/GP_p_form.php 

3. Are Lipid Maps predicted spectra relevant to TOF fragmentation? Some comment is required.

4. Line 334. The authors need to explain more carefully what is meant by "Signal intensities were pseudocolored based on an internal standard reference scale".

5. Figures 3 & 4 have incorrect captions.

6. The statement made on lines 413-415 is questionable from the data provided.

7. Table 1, the authors need to give the search parameters i.e. mass accuracy. When I searched lipidmaps for 734.8 assigned to PC(32:O) I recieved multiple hits. What was the logic behind the selection of PC(32:0). 

Besides this small quires I found this a very interesting article.   

Author Response

Comments

  1. The term "air tight container" was used in many places, what does this actually mean?

We have addressed this definition in Section 2.3.3 TMA Construction, Section 1d.

  1. Line 300. The link does not work. Do the authors mean this link https://www.lipidmaps.org/tools/structuredrawing/GP_p_form.php

Yes, thank you for noting the link error. We have corrected the link in the manuscript.

  1. Are Lipid Maps predicted spectra relevant to TOF fragmentation? Some comment is required.

We have clarified our approach for lipid identification under Section 2.6 MALDI-IMS, #8 and #9.

  1. Line 334. The authors need to explain more carefully what is meant by "Signal intensities were pseudocolored based on an internal standard reference scale".

This point has been clarified.

  1. Figures 3 & 4 have incorrect captions.

Thank you. We have corrected the error by switching the figure captions.

  1. The statement made on lines 413-415 is questionable from the data provided.

We have toned down the final bulleted point in the Discussion (Section 4) since we have not presented evidence along the lines suggested. Such data will be published in future articles.

  1. Table 1, the authors need to give the search parameters i.e. mass accuracy. When I searched lipidmaps for 734.8 assigned to PC(32:O) I received multiple hits. What was the logic behind the selection of PC(32:0).

We used previously published methods to identify lipids detected in the negative and positive modes. The approach used is summarized and referenced in the Table 1 legend and text. We also cite several references used to help identify ambiguous lipids.

Reviewer 4 Report

While the authors have addressed and interesting topic of using frozen tissue microarrays for the analysis of lipid, there are several major concerns with this manuscript that must be addressed prior to publication.

Major Concerns:

1)      The authors claim to have found that formalin fixed and frozen tissue can be used for the analysis of lipids in TMA and indicate that this will make available numerous archived samples.  However, they also indicated that formalin fixed paraffin embedded tissue cannot be used for lipid imaging.  Archived tissue is almost exclusively formalin fixed, paraffin embedded which is room temperature stable and eliminates the need for cryo-storage.  It is extremely rare that tissue would be formalin fixed and then frozen and thus, there would not be many samples that would meet the criteria to be analyzed using this method.

2)      The approach of using formalin fixed brain tissue for lipid imaging is not novel and was shown by Carter, et al. in 2011 (DOI: 10.1007/s13361-011-0227-4).  This paper should be cited in the manuscript.

3)      A major item of concern that is not mentioned at all in this manuscript is the mechanism of formalin fixation and its potential effect on lipid.  Formalin preserves tissue by reacting wit primary amines, mostly in proteins, to form crosslinks.  This stabilizes the tissue for long-term room temperature storage.  However, some classes of lipids (e.g. PE and PS) also contain primary amines and can be come modified and/or crosslinked to other molecules in the tissue.  It should be noted that theses lipid classes may not be suitable for analysis in formalin fixed tissue.

4)      Methods are all self-citations, including review articles.  This requires readers to navigate through several manuscripts to get to the full method.  The methods should ideally be elaborated in this manuscript or provide a direct citation to the full, exact methods used in this study.

5)      A major point of this study was to compare fresh frozen and formalin fixed tissue for lipid mass spectrometry imaging.  However, no spectra are shown to highlight this comparison.  Simply showing one or two ion images does not adequately represent the data and allow for the conclusion to be made.  Spectra from the various conditions should be included for comparison.

6)      To make the overarching conclusion that TMAs as suitable for lipid IMS, the results should be directly comparted to traditional tissue sections to determine similarities and differences.  It is unknown from this study what may be lost when samples are embedded.

Minor Points

1)      No explanation is given as to why gelatin embedding was not evaluated in this study.  Numerous papers have shown gelatin embedding successfully used for imaging of lipids and other small molecules. https://doi.org/10.1007/s13361-017-1851-4 , doi: 10.1016/j.jbc.2021.101139.

2)      In the caption of Figure 1, the authors indicate that sections are 8-16 um in thickness, but on Page 6, line 226 they state that sections are 8-20 um in thickness.  Please make these statements consistent.

3)      Page 6, lines 258-259 – Reference 44 should be added to this step as the washing protocol was first reported in this paper.

4)      Page 7, lines 270-276 – It is not ideal to use different matrices for calibration and analysis, especially one hot (CHCA) and one cold (DHB) as significant changes in laser fluence can affect the mass accuracy.

5)      Page 8, lines 328-330 – The mention of 4% CMC comes out of nowhere for results as it was not mentioned as part of the methods.

6)      Figure 2 caption – The authors mention recipient blocks constructed with ½ mOCT and ½ 2% CMC.  How was this achieved?  From the methods described, theses seem to be separate blocks.  How were they combined?

7)      Page 9, line 351 – The authors mention “identical samples” used for comparison between 2% CMC and mOCT.  Unless a homogenate, tissue-mimetic was uses, different samples will contain different cells and will not be identical.

8)      The captions for Figures 3 and 4 seem to be flipped.

9)      This Figure (currently labeled Figure 3) showing peak intensities of differently lipids in fixed and frozen tissue should not have lines connecting the points as the different species are not related to each other.  These data might be better represented as bar graphs.

10)   When performing data analyses, was histology used to be sure tissue was matched?  With 3 mm punches, there is likely histological variation within each punch.

11)   Table 1 – For the positive mode identification, the [M+K]+ species are likely not a possibility when shown as isobaric with [M+H]+ and [M+Na]+.  PCs with a K cation will show a different headgroup fragment (m/z 163) than those with Na or H (m/z 184) and thus can be differentiated if MS/MS is performed.

12)   Figure 6 – What are the individual species of lipids represented in each bar of the graph?  Are there and trends observed with significance and fatty acid length or number of unsaturations?

Author Response

The authors appreciate this thorough review. Please see the attached point-by-point responses.

Reviewer 5 Report

With the study by Gameiro-Ros et al. the authors address different important topics of investigating tissues at a deep molecular using MALDI-IMS. The authors present different workflows for lipidomic Imaging using tissue micro-arrays.

The study is of big interest for a broad community as the sample preparation, the measurements and the data analysis of MSI for a large sample cohort are complex processes which needs to be investigated in detail to allow for a reproducible and meaningful results.

In general the manuscript is very well written and excellently structured giving a solid knowledge on the topic for a broad readership. I strongly recommend publishing by just putting two minor concerns into account, which are the following:

1.)    Can the authors please comment on the workflow of preparing the TMAs. TMAs are very common in the pathological routine. Is the presented workflow different from these well established ones? What are the specific features for preparing them for MSI?

1.)    Figure 2: I recommend the implementation of the corresponding histology of the tissue, e.g. H&E bright field images to compare the m/z images with the underlying ground truth. Also for Fig. 5 two exemplary cores showing the difference.

Author Response

The authors are very grateful for this excellent and thoughtful review. The recommended revisions have been made. Thanks for the input to help improve the quality of this manuscript.

  • Can the authors please comment on the workflow of preparing the TMAs. TMAs are very common in the pathological routine. Is the presented workflow different from these well established ones? What are the specific features for preparing them for MSI?

The workflow presented is applicable to TMA usage for most histology-based analyses including proteomic, quantitative immunohistochemical staining, and gene expression, but the specific features addressed by the workflow pertain to lipidomics studies which, unlike other analytical approaches, cannot be performed using paraffin-embedded tissue samples.  This point has been reinforced in the manuscript.

2.)    Figure 2: I recommend the implementation of the corresponding histology of the tissue, e.g. H&E bright field images to compare the m/z images with the underlying ground truth. Also for Fig. 5 two exemplary cores showing the difference.

Figures, Brightfield histology-stained sections are included in Figs 2 and Supplementary Figure 1.

Round 2

Reviewer 4 Report

The authors have provided a much-improved revised manuscript.  However, there are still a few points that should be addressed prior to publication.

1) It is still unclear whether nor not MS/MS was used for PIM PC identifications.  If it was, the fragmentation pattern should be able to differentiate [M+H]+ (m/z 184), [M+Na]+ (m/z 146), and [M+K]+ (m/z 163).  Or were multiple species corresponding to all three possibilities detected in the MS/MS spectra?

2) Page 12, lines 452-453 - The authors cite a manuscript stating that "paraffin embedding markedly distorts metabolomics".  While this is true of lipids (which are a lumped under the category of metabolomics) this is not true of all metabolites.  The Buck, et. al (., J Pathol. 2015, 237, 123-132.) showed good correlation between fresh frozen and FFPE tissue for imaging of small polar metabolites.

3) Figure 5 and Figure S1 - Thank you for including these spectra.  However, the spectral quality appear quite poor an many of the individual peaks cannot be observed.  It is not clear if this is an artifact of processing for the version distributed for review or if these have been drawn from a dataset greatly reduced in point.  Please review these figure and try to improve quality if needed.

4) Figure S1 Caption - It is unclear what is being referred to in Figure 4 as this is the principal component analysis.  Should this instead be a reference to Figure 5?

Author Response

The authors appreciate the rich feedback on our manuscript. The overall process has elevated its level of excellence and utility to the scientific community. We have attended to the additional comments and used "track changes" to mark our edits.

1) For this work, we did not use MS/MS to definitively identify PIM-detected PCs. Therefore, we could not differentiate among the possibilities detected in the spectra. We have noted this limitation on Lines 346-348.

2) We have added the reference by Buck, et al and point out that FFPE is suitable for metabolomics but not lipidomics.

3) The figure quality was compressed to keep the file size small, then degraded by the PDF conversion. We have uploaded a high-resolution TIFF format image

4) The S1 caption has been corrected. The relevant manuscript figure is Figure 2.